# A More Sustainable Isocyanide Synthesis from *N*-Substituted Formamides Using Phosphorus Oxychloride in the Presence of Triethylamine as Solvent

**DOI:** 10.3390/molecules27206850

**Published:** 2022-10-13

**Authors:** Sodeeq Aderotimi Salami, Xavier Siwe-Noundou, Rui Werner Maçedo Krause

**Affiliations:** 1Department of Chemistry, Rhodes University, Grahamstown 6140, South Africa; 2Department of Pharmaceutical Sciences, School of Pharmacy, Sefako Makgatho Health Sciences University, Pretoria 0204, South Africa

**Keywords:** *N*-substituted formamides, isocyanides, phosphorus oxychloride, green chemistry

## Abstract

A simple, green, and highly efficient protocol for the synthesis of isocyanides is described. The reaction involves dehydration of formamides with phosphorus oxychloride in the presence of triethylamine as solvent at 0 °C. The product isocyanides were obtained in high to excellent yields in less than 5 min. The method offers several advantages including increased synthesis speed, relatively mild conditions, and rapid access to large numbers of functionalized isocyanides, excellent purity, increased safety, and minimal reaction waste. The new approach of synthesising dehydrative isocyanides from formamides is significantly more environmentally-friendly than prior methods.

## 1. Introduction

Isonitriles, also known as isocyanides or carbylamines, are a diverse class of chemical compounds distinguished by the presence of a terminal isocyano functional group, which causes the unpleasant odour of their volatile representatives [1]. Isocyanides have a peculiar chemistry that allows them to operate as nucleophiles, electrophiles, and even radicals, making them efficient and versatile synthetic reagents [2]. While isonitriles play an important role in a variety of chemical reactions [3], their popularity and significance are strongly linked to some of the most well-known multicomponent methodologies [4], such as the Ugi [5] and Passerini [6] coupling reactions. 

Natural products with an isocyanide functionality (Figure 1) are well known in various kingdoms of living organisms, as intriguing as they may appear. Of course, the origin of such a functional group and the mechanisms that have developed in biosynthetic pathways to allow the formation of isocyanides are crucial questions in biochemistry [7]. Many natural isonitriles have been shown to have potent antibacterial, fungicidal, or anticancer properties and synthetic methods are therefore of great interest [8]. Isonitriles are also employed in the synthesis of heterocyclic systems as versatile building blocks [8]. Recently, a series of terpene isonitriles have been reported to show significant antimalarial activity in vitro [9].

Despite their vast range of applications in organic synthesis, isonitriles have a limited commercial supply and only the simplest, unfunctionalized isocyanide can be purchased. In most cases, access to isonitriles is reliant on laboratory preparation [10]. Current synthetic methods are also limited in scope, cost, and yield. 

Due to the exponential increase in the environmental concerns and regulatory constraints faced in the pharmaceutical and chemical sectors, development of environmentally-benign organic reactions has become a critical and demanding research area in modern organic chemical research [11].

Green chemistry has a lot of potential, not only for reducing by-products, waste, and energy consumption, but also for expanding innovative techniques to new materials while still employing existing technologies [12]. Pharmaceutical and medicinal chemistry may have the highest opportunity for capitalizing on green chemistry technology [13]. From the standpoint of environmentally-acceptable “green chemistry”, a reaction should ideally be carried out in a solvent-free environment with minimal or no side-product generation and maximal atom economy [14].

Isocyanides were discovered 200 years ago and made possible by a nucleophilic substitution reaction of AgCN with alkyl halides [15]. Shortly afterwards, Hoffman discovered dichloro carbene reagents for converting primary amines to isocyanides [16]. The first isocyanide-based multicomponent reaction (IMCR) was found by Passerini, giving isocyanide chemistry a boost [17]. Ugi’s discovery of a two-step technique for reliably synthesizing isocyanides, involving formylation of a primary amine followed by dehydration, was a watershed moment [18].

The Ugi method is still utilized for the vast majority of isocyanide syntheses today, despite the availability of numerous additional isocyanide procedures, such as the enhanced Hoffman approach using phase transfer catalysis (PTK) [4]. The most common method for synthesizing isocyanides is formylation of a primary amine followed by dehydration of the resultant formamide. According to a review of isocyanides, the most practical method of dehydration is with phosphorus oxychloride [19]. 

Many other reagents, including phosgene [20], diphosgene (trichloromethyl carbonochloridate) [21], organic chlorophosphate derivatives [22], XtalFluor-E [23], tosyl chloride [24], and phosphoryl chloride [19,25] have been employed as dehydrating reagents in combination with bases, most commonly tertiary amines, to avoid reduced yields of the products. However, all advances have one or more flaws, such as lengthy, tedious operations, exotic and expensive reagents, enormous waste generation, and prolonged exposure of the chemist to potentially toxic gases, and they frequently have a limited functionalization scope [15,25,26,27]. Despite the fact that phosgene and diphosgene have the highest yields, their use is confined to laboratory-sized work due to their extreme toxicity and difficult handling in the case of phosgene, and their expensive cost in the case of diphosgene [24]. As a result, the phosphoryl chloride approach was utilized in order to achieve yields comparable to phosgene, di-phosgene, and tri-phosgene. The most often used dehydrating agent, phosphorus oxychloride (POCl_3_), creates inorganic phosphate as a by-product, making it preferable to *p*-TsCl, which produces more organic waste [24]. Depending on the reactivity of the formamide, the most typical POCl_3_ method is carried out at low temperatures up to −50 °C. To avoid isocyanide hydrolysis, the workup entails cautious hydrolysis to eliminate any excess dehydrating agent while keeping the pH in the basic range. Finally, distillation, chromatography, or recrystallization must be used to purify the crude isocyanide [19]. Due to the presence of small amounts of acidic materials, this step can also cause significant yield losses. As a result, the traditional POCl_3_-enabled isocyanide synthesis not only necessitates meticulous reaction, workup, and purification conditions, but it is also time-consuming and exposes the chemist to the often smelly isocyanide vapours for an extended period of time.

Furthermore, due to considerable volume of waste generated during the synthesis, traditional isocyanide syntheses cannot be considered sustainable. Wang et al. recently developed a less toxic dehydration reagent based on PPh_3_ and iodine, employing dichloromethane as solvent in the presence of pyridine, which produced high yields of up to 90% within 1 h for mainly aromatic formamides [27]. Shortly after, Porcheddu et al. reported an improvement in the Hoffmann approach towards a more sustainable protocol using mechanochemical activation via ball-milling and significantly reducing the amount of solvent used. As a result, isocyanides with a broad range of aliphatic, benzylic, and aromatic moieties were obtained in 71% yields within 30 min [4]. These few examples demonstrate one of the major drawbacks of this system, namely use of volatile organic solvent, low yield, and long reaction time. Meanwhile, when the three different dehydration reagents (i.e. *p*-toluenesulfonyl chloride (p-TsCl), phosphoryl trichloride (POCl_3_), and the combination of triphenylphosphane (PPh_3_) and iodine were investigated under slightly different experimental conditions for dehydration of formamides by Waibel et al., high yields of isocyanides were generated in the presence of pyridine (organic base), but still the reactions often require constant use of volatile solvent and prolonged reaction times (2 h) [24].

In 2020, Patil et al. published a convenient synthesis protocol for isocyanides towards improving the technique for POCl_3_-based formamide dehydration by avoiding any aqueous workup [19]. The protocol, however, afforded a great structural diversity of isocyanides in high yields and purity within 1 h in the presence of triethylamine. The protocol, however, still relied on the use of volatile organic solvent such as DCM which we tried to avoid for reasons discussed above. There is a need for the development of a rapid, efficient, and more sustainable strategy for the synthesis of Isocyanides. This is because traditional conditions have disadvantages such as toxicity, tedious operations, exotic and expensive reagents, enormous waste generation, prolonged exposure of the chemist to potentially toxic gases, and limited functionalization scope. We hypothesized that the dehydration of formamide to isocyanide can be accomplished without the use of any additional co-solvents, including dichloro-methane, since organic bases like triethylamine and pyridine act as solvents. In view of the limitations of existing methods and as part of our continuous program to develop green reactions, we now report a fast and efficient synthesis of several functionalized isocyanides in the absence of any co-solvents (Figure 2). To the best of our knowledge, no reports on co-solvent-free POCl_3_-based formamides dehydration have been published, and for the first time, we present our findings here.

## 2. Results and Discussion

The dehydration of *N*-(3-bromophenyl)formamide with phosphorus oxychloride, in the presence of triethylamine under co-solvent-free conditions at 0 °C produced 3-bromo-1-isocyanobenzene in excellent yield (Figure 2). The starting point for our experiments was to optimize the reaction conditions, such as solvent and reaction time, for the synthesis of isocyanides with wide structural diversity.

Since the phosphorus oxychloride employed for the dehydration of formamide is a highly reactive reagent, the reaction solvents must be chemically inert, which rules out alcohols, ketones, water, and amines while allowing a variety of acceptable and more sustainable solvents to be employed in the optimization study because some solvents commonly used for the dehydration of *N*-formamides are poisonous based on the greener solvent parameters [28]. Here, our focus is directed towards the development of a more sustainable synthesis approach for isocyanides by significantly reducing the toxicological impact of the synthesis reagent employed as well as minimizing waste (E-factor), two critical factors of overall sustainability.

To achieve sustainable conditions for the above transformation, a series of experiments were carried out. First, we investigated the dehydration of *N*-(3-bromophenyl)formamide with phosphorus oxychloride, in the presence of triethylamine in various co-solvents, and under co-solvent-free conditions. Although the reaction occurs in all the co-solvents tested, dichloromethane and tetrahydrofuran were the best among the tested co-solvents, producing 94% and 72% yields respectively (Table 1).

When the reaction was carried out in diethyl ether, toluene, and acetonitrile, only a small amount of the product was observed producing yields of 37%, 15%, and 56%, respectively (Table 1, entry 1–5). Therefore, DCM was found to be the best organic co-solvent for the current reaction as it provided the desired product in 94% yield within 15 min (Table 1, entry 4). This is unsurprising given that formamides are weakly soluble in most organic solvents. Among the co-solvents examined, dichloromethane was the most effective, which is consistent with literature reports (Table 1, entry 4). However, in the context of sustainability, attention was given to greener alternatives for commonly-used solvents according to solvent selection guides. At this point, we thought of carrying out these reactions under green conditions. To our delight, when the reaction was performed under co-solvent-free conditions, dehydration was completed in 5 min and the product was obtained in 98% yield (Table 1, entry 5), thus lowering the overall environmental impact. 

The best overall results were obtained under co-solvent-free conditions for POCl_3_-based formamides dehydration, which further boosts the reaction’s sustainability because, ideally from the green chemistry point of view, a reaction should be carried out under solvent-free or aqueous conditions. Interestingly, this strategy resulted in an E-factor of 5.5 and a yield of 98%. As a result, this technique proved to be the most sustainable and practicable, and it was employed in further investigations.

However, to generalise the protocol for the dehydration of various functionalized *N*-formamides, we tested various bases, bearing in mind that the dehydration reaction could be accomplished with common organic bases (Table 2, entries 1–4), but not with inorganic bases. Among the tested common organic bases, tertiary amines such as triethylamine proved to be superior (Table 2, entry 2). Therefore, we established optimized conditions for the dehydration of formamides using phosphorus oxychloride for the synthesis of isocyanides. 

However, to generalise the protocol for the dehydration of *N*-formamides for a large number of functionalized isocyanides, the reaction was optimized with respect to temperature and molar ratio. The temperature was reduced to 0 °C, which was found to be sufficient for carrying out the reaction with the highest possible yield of the required product (Table 3, entry 1). It was discovered that using an excess of phosphorus oxychloride was unnecessary because a 1:1 molar ratio of *N*-formamides to phosphorus oxychloride was adequate to produce the desired product (Table 3, entry 1).

In good-to-excellent yields (98–45%), a wide range of *N*-aryl- and *N*-alkyl-substituted formamides were transformed swiftly into the anticipated isonitriles. Substituted *N*-Aryl substrates with an electron-withdrawing or electron-donating group (1–28) on the benzene ring performed satisfactorily. Under these conditions, ether (18–20), halo (1–8), nitro (9–12), ester (22), hydroxyl (12, 27), and nitrile functional group (17) were only slightly affected. Various mono, di, and trisubstituted *N*-phenylformamides (1–34) also yielded good results, indicating that steric effects did not significantly hinder the reaction progress (Table 4). At room temperature, heterocyclic isocyanides, especially those with the isocyano group ortho to the heteroatom, are generally unstable. They tend to undergo further reactions, such as polymerization or cyclization [29]. Our approach, on the other hand, allowed for the synthesis of these compounds with the use of an ice bath and isolation at ambient temperature. In good yields, 2-isocyanothiazole (54%), 4-isocyano antipyrine (68%), and isocyano(pyridin-3-yl)methanone (45%) were isolated. In the cases of 3-isocyano-5-methylisoxazol, 2-(isocyanomethyl)furan, 2-isocyano-3-nitropyridine, 2-isocyano benzyimidazole, 2-isocyanopyridine, 4-isocyanopyridine, 2-(isocyanomethyl)pyridine, and 2-isocyanobenzothiazole, we were unable to isolate the desired product in its purest form. However, the characteristic isocyanide smell of the reaction mixture and the formation of a new spot on the TLC plate clearly indicated their formation. Since our intention was to use these compounds as in situ reactants in various multicomponent reactions, we were not overly concerned at not being able to isolate the pure compounds (vide supra). 

The remaining isocyanides, despite being stable enough to be isolated, decompose rapidly at room temperature and cannot be preserved for any length of time. Isolation of heterocycle-containing isocyanides was complicated by their intrinsic instability. To verify that the herein-obtained isocyanides can be used as in situ reactant in a 3-component Passerini reaction, we chose to proceed with the synthesis of MCR-derived compounds utilizing benzoic acid, benzaldehyde, and 2-isocyanobenzothiazole as in situ reactants in a classical Passerini 3-components reaction (Figure 3). Overall, the findings showed that minor impurities in the isocyanide have no significant impact on the subsequent transformation, suggesting that this approach could be used in other multi-component processes. It is worth noting that these compounds have the potential to undergo the full range of multicomponent reactions for which isocyanides are known, exemplified in, for example Passerini and Ugi reactions, and they have the desirable structural diversity and molecular complexity needed for fine-tuning biological activity.

In order to highlight the superiority of our approach, we calculated the environmental factor (E-factor) and compared it to reported literature protocols for different dehydration reagents including p-toluenesulfonyl chloride (p-TsCl), phosphoryl trichloride (POCl_3_), and the combination of triphenylphosphane (PPh_3_) and iodine, bearing in mind that the lower the E-factor, the better the process’ overall performance. The excessive usage of organic solvent for the dehydration process and aqueous workups adopted during the purification step leads to generation of more aqueous waste for other approaches, thus resulting in a high E-factor (Table 5). 

In addition, while our protocol is solvent free, other approaches leverage on the use of organic solvent about (50–300 mL), particularly DCM, which in turn jeopardized the process’ sustainability (Table 5). Furthermore, the results show that the other approaches require longer reaction times (30 min–2 h) for efficient dehydration than for the present protocol. Based on the number of operations and the experimental simplicity, other approaches required about 7 to 9 workup steps, but our workup consists primarily of a simple silica filtration step followed by solvent evaporation. By avoiding aqueous workup and performing the reaction at higher concentrations under solvent-free conditions, we were able to achieve the lowest E-factor in our procedure resulting into minimal waste generation (Figure 4). In general, the lower the E-factor, the less waste is generated [13]. As a result, our method is more environmentally-friendly than any other known isocyanide synthesis method.

The mechanism of dehydration is illustrated as follows (Figure 5). In the presence of triethylamine, the double bond of the oxygen in the formamide group breaks and protonates the oxygen, which led to the formation of an unstable complex, which, on further elimination of HCl and chloride ions, furnished the anticipated isocyanides. 

## 3. Materials and Methods

All formamides were prepared by the *N*-formylation of various primary amines in the presence of the immobilized sulfuric acid on silica gel (H_2_SO_4_–SiO_2_) as an efficient promoter system. All reactions were performed in refluxing triethyl orthoformate (65 °C) under mild reaction conditions (Figure 6). 

A PerkinElmer Spectrum 100 FT-IR Spectrometer (Valencia, CA, USA) was used for the FT-IR analysis. The IR spectra were obtained by the attenuated total reflection (ATR) method. For each experiment, 16 scans were performed in the frequency range from 650 to 4000 cm^−1^. Melting points of all the compounds were determined using a Kofler hot-stage apparatus and are uncorrected. NMR spectra were recorded on a Bruker AMX400 spectrometer (Rheinstetten, Germany) using CDCl_3_ or DMSO-d_6_ as a solvent with tetramethylsilane used as internal standard. Solvents and chemicals used were of analytical grade, which were purchased from Sigma Aldrich (St. Louis, MO, USA) and used without further purification. The purity determination of the starting materials and reaction monitoring was performed by thin-layer chromatography (TLC) on Merck silica gel G F_254_ plates (Duren, Germany).

### 3.1 Synthesis of Isocyanide from *N*-Formamide

#### General Procedure

To a solution of *N*-formamide (2 mmol) in triethylamine (2 mL), subsequently phosphorus oxychloride (2 mmol) 0.2 mL was added at 0 °C. The reaction mixture was stirred for about 5 min. The progress of the reaction was monitored by TLC. After the completion of the reaction, the reaction mixture was poured directly into a dry packed column. 100% diethyl ether was used as the mobile phase to afford the product in high yield with minimal solvent consumption.

## 4. Conclusions

The dehydration of various *N*-substituted formamides with phosphorus oxychloride afforded smoothly the corresponding isocyanides. Our approach allows synthesis of various functionalized isocyanides in high purity and in almost quantitative yields in solvent-free conditions. This approach enables the in situ synthesis of unstable isocyanides, allowing unstable isocyanides to be used as reactants in a number of processes. As a result, we believe that this method is superior with respect to time, purity, yield, simplicity, safety, and sustainability. Our approach is more environmentally-friendly than the previously reported isocyanide synthesis procedure.

## Figures and Tables

**Figure 1 molecules-27-06850-f001:**
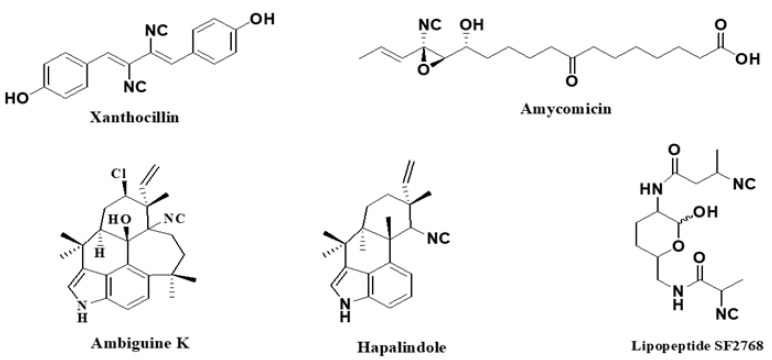
Examples of natural products bearing one or two isocyano functional groups.

**Figure 2 molecules-27-06850-f002:**
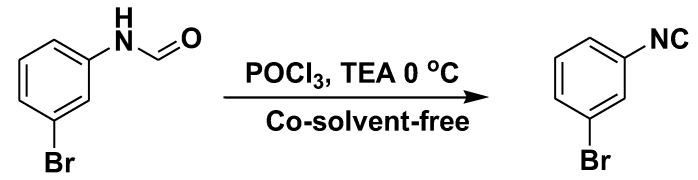
Dehydration of *N*-(3-bromophenyl)formamide to 3-bromo-1-isocyanobenzene.

**Figure 3 molecules-27-06850-f003:**
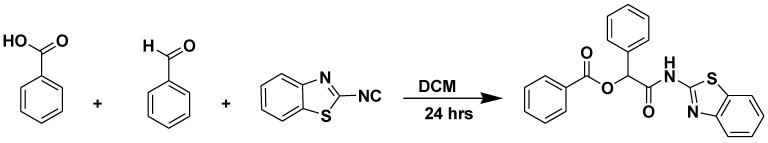
Synthesis of Passerini product using 2-isocyanobenzothiazole generated in situ.

**Figure 4 molecules-27-06850-f004:**
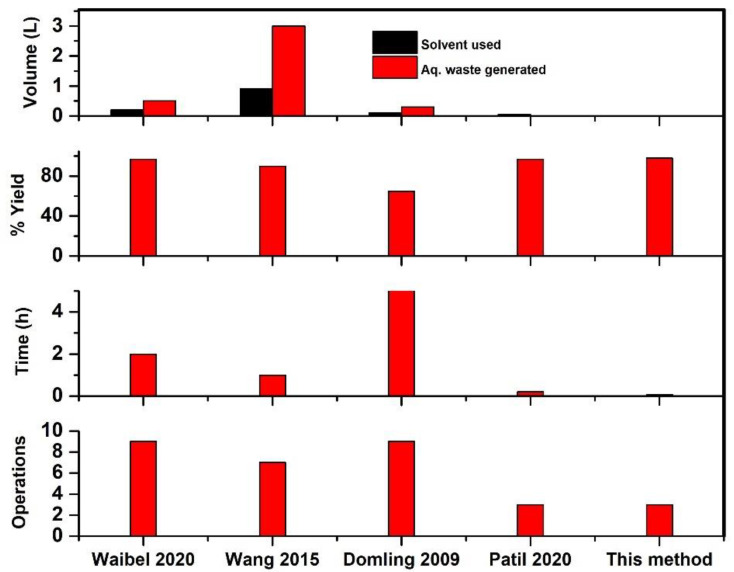
Graphical representation of various parameters for 100 mmol synthesis of various optimized conditions.

**Figure 5 molecules-27-06850-f005:**
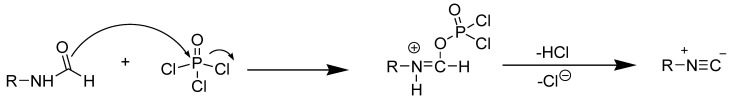
Mechanism of dehydration reaction using phosphorus oxychloride.

**Figure 6 molecules-27-06850-f006:**
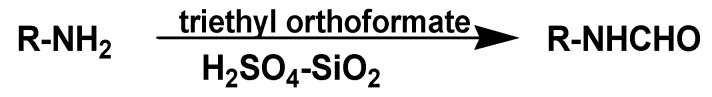
Formamide syntheses.

**Table 1 molecules-27-06850-t001:** Solvent test for the dehydration of *N*-(3-bromophenyl)formamide with POCl_3_ and a base.

Entry	Solvent	Yield %	E-factor	Time
1	THF	72	10.4	1 h
2	Diethyl ether	37	17.2	30 min
2	Toluene	15	20.3	40 min
3	Acetonitrile	56	46.4	1 h
4	DCM	94	8.2	25 min
5	Solvent free	98	5.5	5 min

**Table 2 molecules-27-06850-t002:** Dehydration of *N*-(3-bromophenyl)formamide under different basic conditions.

Entry	Solvent	Base	Time	Yield
1.	Solvent free	Pyridine	20 min	76%
2.	Solvent free	Triethylamine	5 min	98%
3.	Solvent free	Diisopropylamine	25 min	62%
4.	Solvent free	Diisopropyl ethylamine	30 min	59%

**Table 3 molecules-27-06850-t003:** Dehydration of *N*-(3-bromophenyl)formamide with POCl_3_ under different molar and temperature conditions.

Entry	Reaction Condition	Time	Yield
1.	*N*-formamide (1 mmol)/POCl_3_ (1 mmol) 0 °C	5 min	98%
2.	*N*-formamide (1 mmol)/POCl_3_ (1 mmol) r.t.	35 min	51%
3.	*N*-formamide (1 mmol)/POCl_3_ (0.5 mmol) 0 °C	20 min	67%
4.	*N*-formamide (1 mmol)/POCl_3_ (0.5 mmol) r.t.	35 min	28%
5.	*N*-formamide (1 mmol)/POCl_3_ (1.5 mmol) 0 °C	5 min	89%
6.	*N*-formamide (1 mmol)/POCl_3_ (1.5 mmol) r.t.	5 min	84%

r.t. = room temperature.

**Table 4 molecules-27-06850-t004:** Synthesis of isocyanides via formamide dehydration utilizing the optimized reaction conditions with POCl_3_ under solvent free conditions (see above).

Entry	Product	Yield %	Entry	Product	Yield %
1.	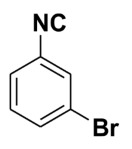	98	2	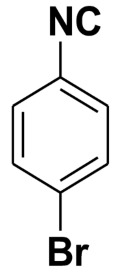	85
3.	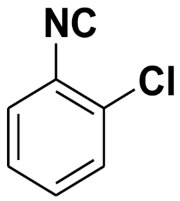	90	4.	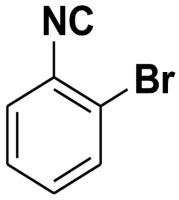	86
5.	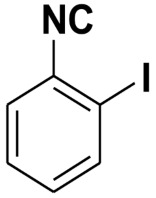	83	6.	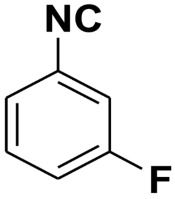	85
7.	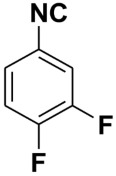	79	8	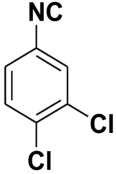	85
9	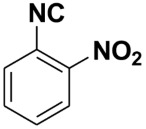	87	10	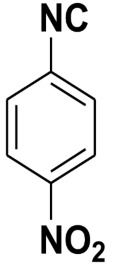	94
11	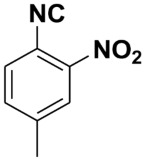	86	12	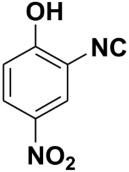	92
13	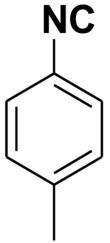	88	14	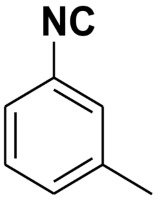	74
15	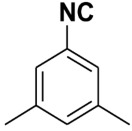	80	16	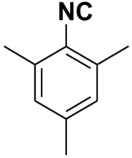	97
17	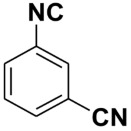	90	18	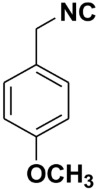	88
19	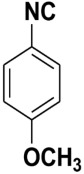	90	20	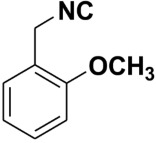	76
21	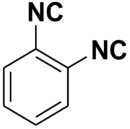	91	22	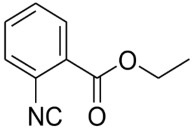	85
23	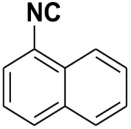	96	24	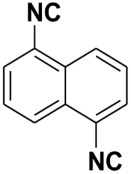	81
25	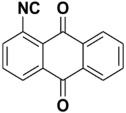	68	26	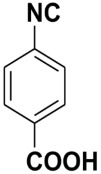	65
27	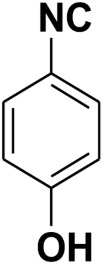	78	28	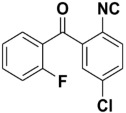	65
29	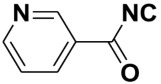	45	30	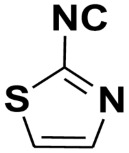	54
31	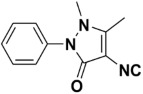	68	32	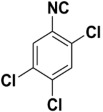	87
33	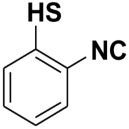	76	34	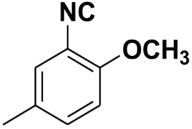	93

**Table 5 molecules-27-06850-t005:** Comparison of various parameters for 100 mmol synthesis of various optimized conditions.

References	Waibel 2020 [18]	Wang 2015 [27]	Domling 2009 [30]	Patil 2020 [19]	This Method
Solvent used (mL)	200	900	100	50	0
Aq. Waste generated (mL)	500	3000	300	0	0
% Yield	97	90	65	97	98
Reaction time (min)	120	60	300	12	5
Number of Operations	9	7	9	3	3
E-Factor	7.41	18.4	50.5	8.4	5.5

## Data Availability

Data from experiments can be accessed from Appendix A.

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
