# Peer review of "A More Sustainable Isocyanide Synthesis from *N*-Substituted Formamides Using Phosphorus Oxychloride in the Presence of Triethylamine as Solvent"

_molecules, 2022, doi:10.3390/molecules27206850_

Round 1
Reviewer 1 Report (Previous Reviewer 1)
Recommendation: The manuscript may be publishable after minor revisions.
In the revised manuscript the authors described a method for the synthesis of isocyanides via dehydration of formamides using phosphorus oxychloride in the presence of triethylamine without using any co-solvent in good to excellent yields. This manuscript is recommended for publication in Molecules after minor revisions:
1. In the Table 4, the “Substrate” word should be changed to “Product”.
2. At the beginning of Table 4, instead of “Synthesized isocyanides”, it should be “Synthesis of isocyanides”
3. In Figure 3., Synthesis of Passerini product using 2-isocyanobenzothiazole, both the reaction sequence is same. So, one of them may be removed.

Author Response
In the Table 4, the “Substrate” word should be changed to “Product”.
Thank you so much for this comment. Adjusted as rightly observed.
At the beginning of Table 4, instead of “Synthesized isocyanides”, it should be “Synthesis of isocyanides”
Thank you for the right observation. Corrected as you observed.
In Figure 3., Synthesis of Passerini product using 2-isocyanobenzothiazole, both the reaction sequence is same. So, one of them may be removed.
Thank you for the right comment. I have adjusted as observed.
Reviewer 2 Report (Previous Reviewer 2)
Although the authors have improved their manuscript, the issues concerning the novelty is still there. Such a routine and incremental work is not suggested to be published in Molecules.
Besides, some structures in of the products are wrongly drawn. The drawings of nitriles and isocyanides should be clarified.
Author Response
Although the authors have improved their manuscript, the issues concerning the novelty is still there. Such a routine and incremental work is not suggested to be published in Molecules.
Thank you very much for the comment. The excessive usage of organic solvent for the dehydration process is a source of worry, despite triethylamine also acting as a solvent. We envisage that there is a need to develop the reaction protocol more sustainably by avoiding this organic solvent. We have shown that the use of co-solvent in the dehydration of formamide to isocyanide is unnecessary as triethylamine act as a solvent. Additionally, the present protocol resulted in the lowest E-factor, which implies that the process performs better than the other protocols. Another advantage of the present protocol is high yields and short reaction time (5 min).
Besides, some structures in of the products are wrongly drawn. The drawings of nitriles and isocyanides should be clarified.
Thanks so much for that observation. This has been corrected.
Reviewer 3 Report (New Reviewer)
Good work - all the experiments are described in detail and spectral data of the products provided is appropriate.
Narration needs to be improvised - some typo & grammatical errors were identified - improvising the English and narration can help the readership.
About the work - In this work researchers have shown that the use of co-solvent is not necessary in this reaction. Although sufficient experimental data is provided - authors can try to shed some lights on mechanism of the reaction, perhaps using the theoretical studies and try to reveal the effect of solvents on this reaction. - This would definitely enhance the impact of this work to many folds.
Author Response
Good work - all the experiments are described in detail and spectral data of the products provided is appropriate.
Thank you very much for the comments.
Narration needs to be improvised - some typo & grammatical errors were identified - improvising the English and narration can help the readership.
Thank you very much for the comment. The article has been checked and updated.
About the work - In this work researchers have shown that the use of co-solvent is not necessary in this reaction. Although sufficient experimental data is provided - authors can try to shed some lights on mechanism of the reaction, perhaps using the theoretical studies and try to reveal the effect of solvents on this reaction. - This would definitely enhance the impact of this work to many folds.
Thank you very much for the comment. The mechanism of the reaction has been added. Although not at a theoretical level.
Round 2
Reviewer 2 Report (Previous Reviewer 2)
Accept
This manuscript is a resubmission of an earlier submission. The following is a list of the peer review reports and author responses from that submission.
Round 1
Reviewer 1 Report
In current work, the authors described a method for the synthesis of isocyanides via dehydration of formamides using phosphorus oxychloride in the presence of triethylamine in good to excellent yields. However, this work is simply a small extension of the paper by Patil and co-workers, P. Patil, M. Ahmadian-Moghaddam, and A. Dömling, Green Chem., 2020, 22, 6902–6911. Therefore, I can’t recommend its publication in Molecules.
Some comments for authors:
1. The authors has used MTBE as reaction solvent and considered this as sustainable approach. However, MTBE is volatile and has low flash point which greatly compromises operational simplicity and environmental safety. Authors should at least provide some references, where MTBE is used in organic reactions and mentioned as green solvent.
2. Cyclopentyl methyl ether (CPME) could be better ether solvent than methyl t-butyl ether (MTBE) as it has higher flash point, boiling point of CPME is 106°C and it has more stability under acidic and basic conditions, Ref., “Cyclopentyl Methyl Ether: An Elective Ecofriendly Ethereal Solvent in Classical and Modern Organic Chemistry” Ugo Azzena, Massimo Carraro, Luisa Pisano, Serena Monticelli, Roberta Bartolotta and Vittorio Pace, ChemSusChem 2019, 12, 40-70.
3. In comparison to Patil’s paper, authors did not explore much on the synthesis of alkyl isocyanides. Only two benzyl isocyanide derivatives, entry 18 and 20, are reported in the table. It would be good to report at least two to three different alkyl isocyanides except benzyl isocyanides.
4. In Supporting information, Material and Methods section detail procedure for N-formylation is absent. If this procedure is previously reported in literature, then author should cite that paper in the reference.
Reviewer 2 Report
The manuscript deals with the dehydration of formamides with phosphorus oxychloride in the presence of triethylamine in MTBE to synthesize isocyanide. The authors insisted on the “Sustainability” of this procedure, for example, by comparing its E-factor with those reported (Table 4). However, overall, the procedure itself is not sustainable. For example, in the experimental section, MTBE (1 ml) and triethylamine (2 ml) were used. From my opinion, triethylamine should be the main solvet while MTBE is only a cosolvent. In addition, the current method, is just a minor modification of previously ones, simply by changing the solvent with MTBE. Thus, it does not exhibit enough novelty for publication in Molecules.
Other questions:
Abstract section, line 10, “at ◦C”what temperature?
Page 4, line 120, “POCl3”
Page 2, lines 70-72, the authors stated that “phosphorus oxychloride (POCl3), creates inorganic phosphate as a by-product, making it preferable to p-TsCl, which produces more organic waste”. However, in the present protocol, organic bases such as TEA was added, they finally became organic waste after the authors’ workup?
The products synthesized are structurally too simple. Can the authors synthesize some more complex isocyanates, for example, those listed in Figure 1?
Figure 2 need to be replaced by a high-resolution version.
Page 12, line 245, “CDCl3 or DMSO-d6”
Reviewer 3 Report
S. A. Salami and R. W. M. Krause report the synthesis of a scope of isocyanide compounds. The data are consistent with the stated conclusions and the provided structural characterizations fully support the purity of the synthesized compounds.
However, the manuscript is sometimes confusing and lacks scientific details. For instance, the model reaction used for exploration of the synthesis conditions is not well-described (associated to table 1) and only the word N-formamides is used instead of the name of the compound itself. In addition, some references are missing (e.g. for the E-factor, see Paul T. Anastas J. Chem. Educ. 2019, 96, 761−765). In addition, I have a major concern about the novelty of the work presented. Even if most of these structures were not described under similar conditions, only minor modifications of an already reported procedure (A. Dömling and coworkers, Green Chem., 2020, 22, 6902-6911) have been performed.
In consequence, I can not recommend the present work for publication.